# Neural Deep Equilibrium Solvers

**Shaojie Bai**
Carnegie Mellon University

**Vladlen Koltun**
Apple

**J. Zico Kolter**
Carnegie Mellon University and
Bosch Center for AI

## Abstract

A deep equilibrium (DEQ) model abandons traditional depth by solving for the fixed point of a single nonlinear layer $f_\theta$. This structure enables decoupling the internal structure of the layer (which controls *representational capacity*) from how the fixed point is actually computed (which impacts *inference-time efficiency*), which is usually via classic techniques such as Broyden's method or Anderson acceleration. In this paper, we show that one can exploit such decoupling and substantially enhance this fixed point computation using a *custom* neural solver. Specifically, our solver uses a parameterized network to both guess an initial value of the optimization and perform iterative updates, in a method that generalizes a learnable form of Anderson acceleration and can be trained end-to-end in an *unsupervised* manner. Such a solution is particularly well suited to the implicit model setting, because inference in these models requires *repeatedly* solving for a fixed point of the *same nonlinear layer* for different inputs, a task at which our network excels. Our experiments show that these neural equilibrium solvers are fast to train (only taking an extra 0.9-1.1% over the original DEQ's training time), require few additional parameters (1-3% of the original model size), yet lead to a 2× speedup in DEQ network inference *without any degradation in accuracy* across numerous domains and tasks.

## 1 Introduction

Recent progress on *implicit* networks, such as Neural ODEs (NODEs) (Chen et al., 2018b; Dupont et al., 2019; Rubanova et al., 2019; Jia & Benson, 2019; Kelly et al., 2020) and deep equilibrium (DEQ) models (Bai et al., 2019; Winston & Kolter, 2020; Kawaguchi, 2021; Bai et al., 2020; Gilton et al., 2021), has motivated this novel class of networks to the forefront of deep learning research. Instead of stacking a series of operators hierarchically, implicit models define their outputs as solutions to nonlinear dynamical systems. For example, DEQ models (which this paper will focus on) define their outputs as fixed points (a.k.a. equilibria) of a layer $f_\theta$ and input $\mathbf{x}$; i.e., output $\mathbf{z}^\star = f_\theta(\mathbf{z}^\star, \mathbf{x})$. Then, in the backward pass, a DEQ implicitly differentiates through the final fixed point $\mathbf{z}^\star$ (Krantz & Parks, 2012; Bai et al., 2019; Fung et al., 2021), regardless of how forward pass is computed in the first place. Such insulated forward and backward passes enable an equilibrium model to leverage arbitrary black-box solvers to reach the fixed points without storing intermediate activations, thus consuming *constant training memory*. Recent works have successfully applied the DEQ framework on high-dimensional tasks such as language modeling (Merity et al., 2017) and semantic segmentation (Cordts et al., 2016), with performance competitive with architectures like Transformers (Vaswani et al., 2017; Dai et al., 2019).

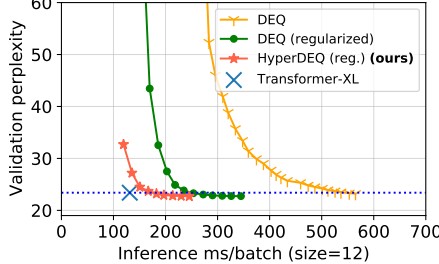

Figure 1: Pareto curves of the same DEQ with different solvers on WikiText-103 language modeling (on 1 GPU).

However, it is also well-known that these implicit models are slow, which is (arguably) their *single most limiting drawback* compared to traditional feedforward models (Duvenaud et al., 2020; Dupont et al., 2019; Bai et al., 2021). For example, Neural ODEs could take well over 100 forward solver iterations (i.e., evaluations of $f_\theta$) even on MNIST classification; DEQs can scale to realistic tasks, but

the overhead of fixed-point solvers is magnified by the task scales, rendering the model 3-6× slower than state-of-the-art (SOTA) explicit networks (Vaswani et al., 2017; Wang et al., 2020) at inference.

*Can we make equilibrium models faster by taking advantage of their implicitness*? One benefit of DEQ's formulation is the fact that they decouple the ***representational capacity*** (determined by $f_\theta$) and ***forward computation*** (controlled by the solver), which is not possible in any explicit model (e.g., ResNet-101 (He et al., 2016)). Hence, given a trained DEQ, one can trade off inference time and the accuracy of the estimated fixed point by simply reducing the number of solver iterations. This yields a speed/accuracy trade-off curve, as shown in Fig. 1. However, this trade-off (i.e., movements *along* the pareto curves) can be highly risky: as we gradually increase inference speed by compromising the quality of fixed point estimates, model accuracy also degrades drastically.

In this work, we show that we can shift the DEQ speed/accuracy trade-off curve by exploiting such decoupling to customize the fixed-point solving. Prior work on equilibrium models relies on classic solvers, which are manually designed and generic (e.g., Broyden's Method (Broyden, 1965)). We propose a tiny, learnable, and *content-aware* solver module that is automatically *customized* to a specific DEQ. Our *hypersolver* consists of two parts. First, we introduce a learned initializer that estimates a good starting point for the optimization. Second, we introduce a generalized parameterized version of Anderson mixing (Anderson, 1965) that learns the iterative updates as an input-dependent temporal process. Overall, the hypersolver consumes a tiny amount of parameters. Since $f_\theta$ is frozen when the hypersolver is trained, the training is very fast and does not compromise generalization.

Our experiments apply this approach to diverse domains with large datasets: WikiText-103 language modeling (Merity et al., 2017), ImageNet classification (Deng et al., 2009), and Cityscapes segmentation with megapixel images (Cordts et al., 2016). Our results suggest that neural deep equilibrium solvers add little overhead to training (only taking an extra 0.9-1.1% over the original DEQ's training time), are extremely compact (about 1-3% of the DEQ's model size), and lead to a consistent and universal 1.6-2× acceleration of inference with no compromise in accuracy. Overall, we believe this paper achieves two major objectives, both vital for the quickly growing community studying implicit models: first, we advance these large-scale implicit models to a much more practical level across architectures (e.g., almost as fast as Transformers); and second, we formally bring up and exploit this valuable notion of how implicit layers decouple representational capacity and forward computation, opening a new door to significantly advancing the agenda of deploying implicit models in practice.

## 2 RELATED WORK

**Deep Implicit Models.** Recent research on models without a prescribed computation graph or hierarchical stacking led to a new class of deep learning models where the output is defined as the solution of nonlinear systems (Duvenaud et al., 2020; Amos & Kolter, 2017; Chen et al., 2018b; Wang et al., 2019; El Ghaoui et al., 2019; Bai et al., 2019; 2020; Gould et al., 2019; Gu et al., 2020; Wang et al., 2020). Neural ODEs (NODEs) (Chen et al., 2018b; Dupont et al., 2019), for example, model infintesimal steps of a residual layer $f_\theta$ by solving an initial value problem (IVP) (Coddington & Levinson, 1955) parameterized by this layer; i.e. $\frac{\partial \mathbf{z}}{\partial t} = f_\theta(\mathbf{z}(t), t), \mathbf{z}(0) = \mathbf{x}, t = 0, \ldots, T$. Deep equilibrium (DEQ) models (Bai et al., 2019; Winston & Kolter, 2020) seek to directly solve for a "fixed-point" representation corresponding to a (not necessarily residual) layer $f_\theta$ and input $\mathbf{x}$; i.e. $\mathbf{z}^\star = f_\theta(\mathbf{z}^\star, \mathbf{x})$. Implicit models are appealing in part due to their analytical backward passes (e.g., adjoint method or implicit differentiation) that only depend on the final output, which can dramatically reduce memory consumption during training.

**Regularizing Implicit Models.** Implicit models are known to be slow during training and inference. To address this, recent works have developed certain regularization methods that encourage these models to be more stable and thus easier to solve. For NODEs, Dupont et al. (2019) augment the neural ODE hidden state; Grathwohl et al. (2019) use spectral normalization (Miyato et al., 2018) to stabilize the NODE dynamics; Kelly et al. (2020) regularize higher-order time derivatives of the ODE system. For DEQs, Winston & Kolter (2020) propose a parameterization of $f_\theta$ that guarantees stability of DEQ models (i.e., unique fixed point). Fung et al. (2021) show that one can simplify the implicit differentiation of Lipschitz DEQs (Revay et al., 2020) to accelerate the backward pass. Bai et al. (2021) summarize DEQ stability issues and propose to address them by regularizing the Jacobian matrices of equilibrium layers. In comparison, our work focuses on the solver rather than the layer $f_\theta$, and is orthogonal and complementary to regularization methods.

**Improving Implicit Model Solvers.** Of particular relevance to our work are recent advances in the Neural ODE literature that improve the ODE flow solver. Poli et al. (2020) introduce a Neural ODE formulation that adds a learnable residual fitting step to the original solver steps, aiming to approximate the higher-order terms of canonical ODE solvers (e.g., Euler's method) on each solution checkpoint along the ODE path. Another recent work (Kidger et al., 2021) focuses on improving the adjoint method by replacing the usual L2 norm with a more flexible seminorm to make the NODE backward solver faster. To the best of our knowledge, no such solver improvement has been explored in the equilibrium model context. Unlike Neural ODEs, DEQs do not use ODE solvers and do not have unique & well-defined trajectories to the solution (even if one starts at the same initial point $\mathbf{z}^{[0]}$). Our work is the first to propose a neural fixed-point solver for equilibrium models.

**Learning to Optimize/Learn.** An important line of work has explored learnable optimization methods. Li & Malik (2016; 2017) propose to use reinforcement learning (guided policy search) to learn a new *generic* unconstrained continuous optimization algorithm, where the training set consists of numerous randomly generated objective functions. Andrychowicz et al. (2016) introduce the "learning to learn" (L2L) framework, where a gradient update rule for the parameters is learned by an LSTM with a pre-defined horizon $T$ of parameter update steps. However, such approaches (Andrychowicz et al., 2016; Chen et al., 2017; Wichrowska et al., 2017; Ravi & Larochelle, 2016) have had some difficulty in generalizing to larger tasks due to the need to unroll for a large $T$ (e.g., 128 (Andrychowicz et al., 2016)). Our work is related to these prior efforts in L2L, but differs in important ways. First, the L2L framework aims to learn a learning algorithm that will be applied to multiple models and tasks, while we aim to fit the nonlinear dynamics of a specific implicit model. Second, the optimization we tackle is not on the parameter space, but on the hidden unit space; this means that the RNN optimizer used in L2L would not work here, because the fixed points themselves can be of variable sizes at test time (e.g., sequence lengths, image sizes). Third, while L2L methods cannot know a priori what a good "initial guess" of optimal parameters may be, we show that it is possible and reasonable to infer this in the hidden unit space with implicit models. Concurrent to our work, Venkataraman & Amos (2021) studies an RNN-based learnable fixed-point acceleration scheme specifically in the application of convex cone programming.

## 3    BACKGROUND: EQUILIBRIUM MODELS AND FIXED-POINT SOLVERS

**Deep Equilibrium Models.** Given a layer (usually a shallow block; e.g., self-attention (Vaswani et al., 2017)) $f_\theta$ and an input $\mathbf{x}$, a DEQ model aims to solve for an "infinite-level" feature representation without actually stacking the $f_\theta$ layer infinite times. Instead, we can solve directly for the fixed point $\mathbf{z}^\star$ of the system:

$$g_\theta(\mathbf{z}^\star, \mathbf{x}) := f_\theta(\mathbf{z}^\star, \mathbf{x}) - \mathbf{z}^\star = 0.$$

The fixed point can be estimated by quasi-Newton (or Newton's) methods, which provide superlinear (or even quadratic) convergence (Broyden, 1965; Anderson, 1965). Subsequently, in the backward pass, one can implicitly differentiate through the equilibrium point, even without knowledge of how it is estimated, and produce gradients with respect to the model parameters $\theta$ by solving a Jacobian-based linear equation:

$$\frac{\partial \ell}{\partial \theta} = \frac{\partial \ell}{\partial \mathbf{z}^\star} \bigg( I - \underbrace{\frac{\partial f_\theta(\mathbf{z}^\star, \mathbf{x})}{\partial \mathbf{z}^\star}}_{\text{Jacobian of } f_\theta} \bigg)^{-1} \frac{\partial f_\theta(\mathbf{z}^\star, \mathbf{x})}{\partial \theta} = -\frac{\partial \ell}{\partial \mathbf{z}^\star} J_g(\mathbf{z}^\star)^{-1} \frac{\partial f_\theta(\mathbf{z}^\star, \mathbf{x})}{\partial \theta}. \tag{1}$$

The most important message from Eq. equation 1 is that the backward pass can be computed with merely the knowledge of $\mathbf{z}^\star$, irrespective of how it is found. More recently, Fung et al. (2021) prove the feasibility of directly replacing $J_g(\mathbf{z}^\star)$ with $-I$ (i.e., Jacobian-free backward pass), which significantly accelerates training.

**Fixed-point Solvers for DEQs.** Prior works have explored a number of techniques for finding the fixed points of DEQs. For example, Bai et al. (2019; 2020); Lu et al. (2021) used Broyden's method (Broyden, 1965), the memory consumption of which grows linearly with the number of iterations since all low-rank updates are stored. Other recent work (Duvenaud et al., 2020; Gilton et al., 2021) shifted to Anderson acceleration (AA) (Anderson, 1965), a lightweight solver that is provably equivalent to a multi-secant quasi-Newton method (Fang & Saad, 2009). We briefly introduce AA here, since our approach will use it as the starting point.

---

**Algorithm 1** Anderson acceleration (AA) prototype (with parameter $\beta$ and $m$)

---

1: **Input:** initial point $z^{[0]} \in \mathbb{R}^n$, fixed-point function $f_\theta : \mathbb{R}^n \to \mathbb{R}^n$, max storage size $m$
2: **for** $k = 0, \dots, K$ **do**
3:     1) Set $m_k = \min\{m, k\}$
4:     2) Compute weights $\alpha_i^k$ for the **past** $m_k$ **Anderson steps** s.t. $\sum_{i=0}^{m_k} \alpha_i^k = 1$.
5:     3) $z^{[k+1]} = \beta \sum_{i=0}^{m_k} \alpha_i^k f_\theta(z^{[k-m_k+i]}) + (1-\beta) \sum_{i=0}^{m_k} \alpha_i^k z^{[k-m_k+i]}$     (AA_update step)
6: **end for**

---

Prototype algorithm 1 illustrates the main idea of Anderson acceleration: we maintain a size-$m$ storage of the most recent steps, and update the iteration as a normalized linear combination of these steps with weights $\alpha_i$ (step 3). In the canonical AA algorithm, the weights are computed in a greedy manner at each step to minimize the linear combination:

$$\alpha^k = \arg \min_{\alpha \in \mathbb{R}^{m_k+1}} \|G^{[k]} \alpha\|_2, \text{ s.t. } \mathbf{1}^\top \alpha = 1, \tag{2}$$

where $G^{[k]} = [g_\theta(\mathbf{z}^{[k-m_k]}) \ \dots \ g_\theta(\mathbf{z}^{[k]})]$ are the past (up to $m + 1$) residuals; typically, $\beta = 1$ and $m \leq 5$. Eq. equation 2 can be solved by a least-squares method. In all prior works with DEQs (Bai et al., 2019; 2020; Winston & Kolter, 2020; Revay et al., 2020; Fung et al., 2021; Lu et al., 2021), the fixed point iteration starts with an initial $\mathbf{z}^{[0]}$ that is either 0 or a random sample from $\mathcal{N}(0, I)$.

## 4 NEURAL DEEP EQUILIBRIUM SOLVERS

While classic fixed-point estimation algorithms, as presented in Section 3, already work well, they are generic and make minimal assumptions about the specific problem being solved. For example, while multiple papers in optimization literature have acknowledged that tuning $m$ (and $m_k$) as well as varying $\beta = (\beta_k)_{k=0,\dots,K}$ *for each Anderson iteration* $k$ could accelerate AA's convergence to the fixed point (Anderson, 1965; Fang & Saad, 2009; Walker & Ni, 2011), this is rarely considered in practice because it's unclear what schedule should be applied to these parameters.

We propose to make fixed-point solvers for DEQ models learnable and content-based, which is made possible by the unique properties of implicit models. First, unlike generic problems, the nonlinear system for each DEQ is uniquely defined by the input $\mathbf{x}$ (e.g., an image, etc.): $\mathbf{z}^\star(\mathbf{x}) = \mathbf{z}^\star = f_\theta(\mathbf{z}^\star, \mathbf{x})$. This opens the door to learning to make an informed initial guess, followed by content-based iterative updates in the solver. Second, due to implicit models' disentanglement of representation capacity with forward computation, our goal of improving solvers is decoupled from the original learning goal of the DEQ model itself (i.e., the solver is *not* aware of the original task, such as to predict the class of an image). Hence, we are able to train this neural solver in a *lightweight* and *unsupervised* manner, directly with the help of groundtruth fixed-point solutions (see below).

### 4.1 GENERAL FORMULATION

For a given DEQ layer $f_\theta$ and (possibly random) input $\mathbf{x}$, we assume access to its exact fixed point $\mathbf{z}^\star = \mathbf{z}^\star(\mathbf{x}) = f_\theta(\mathbf{z}^\star, \mathbf{x})$, which can be obtained by taking a classic solver (e.g., Broyden's method) and running it for as many iterations as needed (e.g., 100 steps) to a high level of precision.

The overall structure of the hypersolver is shown in Fig. 2. We use a tiny neural network parameterized by $\omega = \{\phi, \xi\}$ (explained below) to learn the initialization and iterative solving process, and unroll the learnable solver for some $K$ steps to yield a prediction $\mathbf{z}^{[K]}(\mathbf{x})$. To train this neural solver, we minimize an objective $\mathcal{L}(\omega, K)$ (discussed in Sec. 4.2) by backpropagating through this $K$-step *temporal* process (Mozer, 1989; Robinson & Fallside, 1987). The original DEQ parameters $\theta$ are frozen, and only the hypersolver parameters $\omega$ are trained here. We also do not need the groundtruth label $y$ (e.g., the class of an image) that corresponds to input $\mathbf{x}$, which means these neural equilibrium solvers can also be fine-tuned on the fly after deployment, at inference time.

**Initializer.** The initial values can have a significant impact on the optimization process and its convergence speed. We propose to make an input-based guess with a tiny network $h_\phi$: $\mathbf{z}^{[0]} = h_\phi(\mathbf{x})$, where $\phi$ are the parameters. Note that the goal of the initializer is not to solve the underlying problem

---

**Algorithm 2** HyperAnderson Iterations (parameterized parts highlighted in color)

---

1: **Input:** initial point $\mathbf{z}^{[0]} = h_\phi(\mathbf{x}) \in \mathbb{R}^n$, (frozen) layer $f_\theta$, storage $G = \mathbf{0} \in \mathbb{R}^{(m+1)\times n}$ with size $m+1$, HyperAnderson network $s_\xi$.
2: Define $g_\theta(\mathbf{z}) = f_\theta(\mathbf{z}) - \mathbf{z}$. Set $G[0] = g_\theta(\mathbf{z}^{[0]})$.
3: **for** $k = 0, \ldots, K$ **do**
4:     Set $m_k = \min\{m, k\}$ and $G^{[k]} = G[0:(m_k+1)] \in \mathbb{R}^{(m_k+1)\times n}$
5:     Compute $\hat{\alpha}^k, \beta_k = s_\xi(G^{[k]})$, where $\hat{\alpha}^k = (\hat{\alpha}_0^k, \ldots, \hat{\alpha}_{m_k}^k) \in \mathbb{R}^{(m_k+1)}$
6:     $\alpha^k = \hat{\alpha}^k + \frac{(1 - \mathbf{1}^\top \hat{\alpha}^k)}{m_k+1} \cdot \mathbf{1}$                                     (normalization step)
7:     $\mathbf{z}^{[k+1]} = \beta_k \cdot \mathbf{1}^\top G^{[k]} + \sum_{i=0}^{m_k} \alpha_i^k \mathbf{z}^{[k-m_k+i]}$     (same AA_update as in Alg. 1, simplified)
8:     Update $G = \texttt{concat}(G[1:], [g_\theta(\mathbf{z}^{[k+1]})])$
9: **end for**
10: **Return** $\mathbf{z}^{[k+1]}$

---

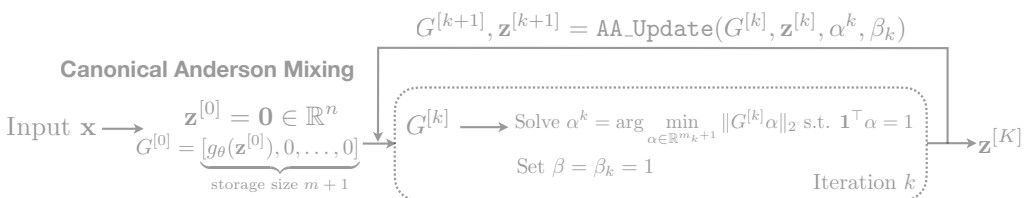

(a) The original generic Anderson solver

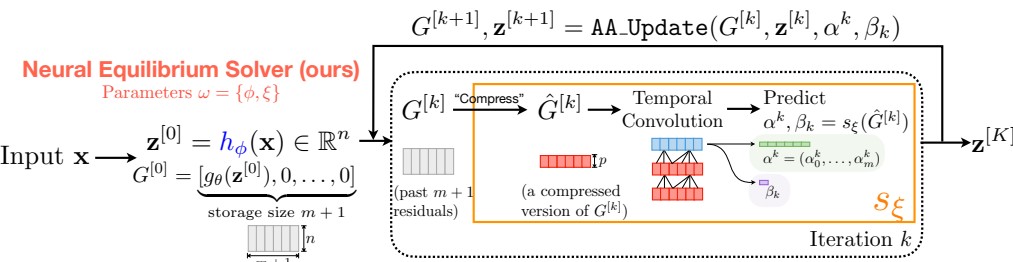

(b) Our proposed (tiny but learnable) HyperAnderson solver

Figure 2: 2a: The canonical Anderson solver is based on a local least-squares solution at each iteration, with $\beta = \beta_k$ set to a constant. 2b: Our neural fixed-point solver provides a *better initial guess* $\mathbf{z}^{[0]}$ and *learnable iterative updates*.

at all (e.g., to classify an image; we don't even need the groundtruth label $y$), but only to yield a quick initial estimate. For example, in language modeling, where $\mathbf{x} \in \mathbb{R}^{T\times d}$ is a length-$T$ sequence, we set

$$h_\phi(\mathbf{x}) = \text{ReLU}(\texttt{Conv1d}_{k=3}(\mathbf{x}))W \ , \text{ where } \texttt{Conv1d}_{k=3} : \mathbb{R}^{T\times d} \to \mathbb{R}^{T\times p} \tag{3}$$

and where $W \in \mathbb{R}^{p\times q}$, with $q$ being the dimension of the fixed point of a single token. We set $p$ to be very small (e.g., 100), so that $h_\phi$ is tiny and fast. Note that this 1-layer initializer by itself has very low expressivity and is usually a poor model for the original task, as we verify in Sec. 5.3.

**HyperAnderson Iterations.** We further parameterize the setting of $\beta_k$ and $\alpha_i^k$ while following the AA prototype outlined in Alg. 1. In lieu of setting Eq. 2 for $\alpha$ to a least-squares solution over the past few residuals $G$, we make both $\alpha \in \mathbb{R}^{(m_k+1)}$ and $\beta \in \mathbb{R}$ explicit learnable functions of $G$ with a neural network $s_\xi(G) : \mathbb{R}^{(m_k+1)\times n} \to (\mathbb{R}^{(m_k+1)} \times \mathbb{R})$; see Alg. 2.

A challenge here is that $n$ (the dimension of $\mathbf{z}^\star$) is typically large in practice, as it is affected by the scale of the input (e.g., in DEQ sequence models (Bai et al., 2019), $n$ is over $1.5 \cdot 10^5$ on a *single* textual sequence of length 200). This makes $s_\xi$ map from an extremely high-dimensional space to a low-dimensional space (e.g., $m = 5$). To keep $s_\xi$ fast, small, and applicable to inputs of varying dimensionalities (e.g., sequence length or image size), we propose to first compress each $g_\theta(\mathbf{z}^{[k]})$ to form a smaller yet still representative version $\hat{G}^{[k]}$ of $G^{[k]} = [g_\theta(\mathbf{z}^{[k-m_k]}), \ldots, g_\theta(\mathbf{z}^{[k]})]$. For example, when each $g_\theta(\mathbf{z}^{[k]})$ is a image feature map residual of dimension $n = C \times H \times W$, we can

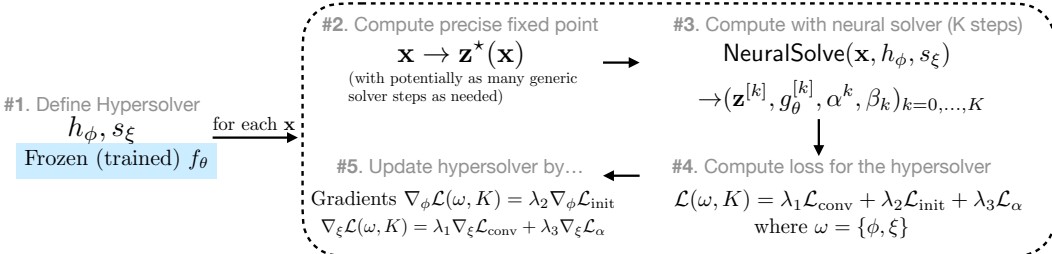

Figure 3: The training procedure of the neural deep equilibrium solver. With a given $f_\theta$ and input $\mathbf{x}$, we optimize the hypersolver parameters $\omega = \{\phi, \xi\}$ via losses applied on the HyperAnderson iterations and the initializer (see Sec. 4.2).

perform global pooling to form a $C$-dimensional vector $\text{Pool}(g_\theta(\mathbf{z}^{[k]}))$ as its compressed version:

$$\hat{G}^{[k]} = [\text{Pool}(g_\theta(\mathbf{z}^{[k-m_k]})), \dots, \text{Pool}(g_\theta(\mathbf{z}^{[k]}))] \in \mathbb{R}^{C \times (m_k+1)}, \quad \text{and predict } \alpha^k, \beta_k = s_\xi(\hat{G}^{[k]}) \tag{4}$$

Once we have this representative collection $\hat{G}^{[k]}$, we treat it as a mini time-series of length $(m_k + 1)$ that encodes the latest estimates of the fixed point. We then apply a 2-layer temporal convolution (van den Oord et al., 2016) to learn to predict: 1) a relative weight $\alpha_i^k$ for each of these past residuals $i \in [m_k]$; and 2) the HyperAnderson mixing coefficient $\beta_k$ for the current iteration. Therefore, $s_\xi$ shall gradually learn to adjust these parameters $\alpha$ and $\beta$ in light of the previous hypersolver steps, and receive gradients from later iterations. We explain the detailed design choices of $s_\xi$ in Appendix B, while noting that it still *completely captures* the AA prototype (see Alg. 1).

## 4.2 TRAINING THE NEURAL EQUILIBRIUM SOLVERS

One benefit of training hypersolvers on implicit models is that they can be trained in an *unsupervised* manner via $\mathbf{z}^\star(\mathbf{x})$, which a slower classic method can provide as many as needed, and for any given (possibly even random) input tensor $\mathbf{x}$. Moreover, unlike NODE solvers (Chen et al., 2018b; Poli et al., 2020), a DEQ model does not have a unique trajectory and thus its hypersolvers do not need trajectory fitting at all. All that we need is to drive everything to be as close to $\mathbf{z}^\star$ as possible. As an example, a neural solver could learn to sacrifice progress in earlier iterations if it subsequently converges to the equilibrium faster. Formally, given a hypersolver $\{h_\phi, s_\xi\}$ that yields a set of states $(\mathbf{z}^{[k]}, G^{[k]}, \alpha^k, \beta_k)_{k=0,\dots,K}$ (recall $\mathbf{z}^{[0]} = h_\phi(\mathbf{x})$), we introduce 3 objectives for its training.

**Fixed-point Convergence Loss.** The first loss aims to encourage convergence at all intermediate estimates $[\mathbf{z}^{[k]}]_{k=1,\dots,K}$ of the HyperAnderson iterations: $\mathcal{L}_{\text{conv}} = \sum_{k=1}^K w_k \|\mathbf{z}^{[k]} - \mathbf{z}^\star\|_2$, where $w_k$ is the weight for the loss from iteration $k$ such that $\sum_{k=1}^K w_k = 1$. We set $w_k$ to be monotonically increasing with $k$ such that later iterations apply a heavier penalty for deviation from the fixed point.

**Initializer Loss.** We also train the initializer by maximizing the proximity of the initial guess to the fixed point: $\mathcal{L}_{\text{init}} = \|h_\phi(\mathbf{x}) - \mathbf{z}^\star\|_2$, We separate this objective from $\mathcal{L}_{\text{conv}}$ since the initialization is predicted directly from the input $\mathbf{x}$ and does not go through HyperAnderson updates.

**Alpha Loss.** Although we replace the generic Anderson solver (Anderson, 1965) in terms of how $\alpha^k, \beta_k$ are computed in each iteration, we empirically found it still beneficial to guide the hypersolvers' prediction of $\alpha$ with an auxiliary loss especially at the start of the training: $\mathcal{L}_\alpha = \sum_{k=0}^K \|G^{[k]} \alpha^k\|_2$. In practice, we gradually decay the weight of this loss to 0 as training progresses. We summarize the complete training procedure of a neural solver on top of a DEQ in Fig. 3.

## 4.3 DISCUSSION

**Complexity of hypersolver.** Note that $f_\theta$ remains frozen during hypersolver training. This means that for a given DEQ model $f_\theta$ and input $\mathbf{x}$, the fixed point $\mathbf{z}^\star(\mathbf{x}) = f_\theta(\mathbf{z}^\star; \mathbf{x})$ also remains the same – we are just trying to learn to find it faster, with a limited $K$-iteration budget. Moreover, we designed the initializer $h_\phi$ and HyperAnderson network $s_\xi$ to be intentionally simple (e.g, 1 layer with few hidden units), so that *each hypersolver step is even faster than the original Anderson step*, whose main computational overhead occurs in solving the constrained optimization in Eq. 2.

These points also highlight the difference between the neural solver and techniques such as model compression (Han et al., 2015) or distillation (Hinton et al., 2015), where a pruned/smaller (but still

representationally rich) model is trained to match the output and performance of a larger model. Specifically, in our case, as the fixed point $\mathbf{z}^\star$ is determined solely by $f_\theta$ and $\mathbf{x}$, the hypersolver itself does not have much representational capacity, since its only goal is to produce an "educated" initial guess and learnable iterations to facilitate the optimization process. E.g., the 1-layer Conv1d-based initializer Sec. 4.1 would be a bad language model by itself since it is tiny and only sees the past 2 tokens (see Sec. 5.3 for empirical evidence), yet this limited capacity and context turn out sufficient to guide and substantially improve the solver.

**Training hypersolver via BPTT.** While a generic Anderson solver computes $\alpha^k$ by optimizing locally with $G^{[k]}$, backpropagating through the HyperAnderson steps ensures that the iterative update network $s_\xi$ can receive gradient and learn from later iterations. This is appealing because, arguably, only the output of the $K^{\text{th}}$ iteration matters in the end. Indeed, we empirically verify via ablation studies in Sec. 5 that such learned $\alpha$ and $\beta$ predictors already significantly accelerate the convergence process even without the presence of the initializer. Note that as DEQ models' $f_\theta$ layer is typically richly parameterized, the backpropagation-through-time (BPTT) might consume a lot of memory. To limit memory consumption, we use small batch sizes for hypersolver training. (This does not affect the training of the DEQ model itself, which is separate.) We have observed that hypersolver training is highly effective with small batch sizes, as reported in Sec. 5 and App. A. As an alternative solution, since these hypersolvers are very fast to train in practice, one could also use methods such as gradient checkpointing (Chen et al., 2016).

**Complementarity with DEQ regularizations.** Besides tiny size and fast training, the value and usefulness of neural equilibrium solvers are highlighted by how DEQ models decouple representational capacity and forward solver choice. In particular, our method is orthogonal to prior work that accelerates DEQ models by structural regularization of $f_\theta$ (Winston & Kolter, 2020; Revay et al., 2020; Bai et al., 2021) or approximating the Jacobian of $f_\theta$ in the backward pass (Fung et al., 2021). In Sec. 5, we show evidence that our method (which is solver-based) integrates well with regularization approaches (which are $f_\theta$-based) and yields broad improvements compared to canonical solvers (e.g., Broyden or Anderson methods) regardless of how $f_\theta$ was trained or what structure it uses.

## 5 EXPERIMENTS

In this section, we verify the various benefits of exploiting neural solvers in implicit models. Specifically, as our goal is to show the superiority of the learnable solvers over generic solvers on both performance and efficiency aspects, we compare the movement of the *entire* speed/accuracy pareto curve rather than a single point on the curve. To achieve this purpose, we study the hypersolver on some of the largest-scale experiments that DEQs have been used on: WikiText-103 language modeling (Merity et al., 2017), ImageNet classification (Deng et al., 2009), and Cityscapes semantic segmentation with megapixel images (Cordts et al., 2016). Overall, we show that: 1) neural solvers bring universal improvement over generic solvers on DEQ models in all scenarios, with a typically 1.6-2× speedup at inference and no loss in performance (i.e., the new pareto curves *strictly dominates* old ones); 2) these hypersolvers can be trained very quickly; and 3) these methods complement prior methods such as regularizations on $f_\theta$ to bring these implicit models to a new competitive level. At the end of this section, we also conduct extensive ablative studies on the design of the hypersolver.

Note that since the neural solver training is independent of the DEQ training, we do not need to train the actual DEQ model $f_\theta$ itself (but could instead directly work on top of a pre-trained DEQ). Therefore, the major hyperparameters in our setting are only the relative weights of the loss objectives (see Sec. 4.2 and Appendix A). We also clarify that the use of hypersolver does implicitly assume local stability around $\mathbf{z}^\star$ for convergence – which we find almost always holds empirically, and can be regularized for (Bai et al., 2021). Code is available at `https://github.com/locuslab/deq`.

### 5.1 LARGE-SCALE EXPERIMENTS ON VISION AND LANGUAGE TASKS

To evaluate the the neural deep equilibrium solvers, we apply them on three largest-scale and highest-dimensional tasks the implicit models have ever been applied on, across the vision and language modalities. In contrast to prior works (Chen et al., 2018b; Winston & Kolter, 2020; Bai et al., 2021) that measure the number of function evaluations (NFEs), we directly measure wall-clock inference speed under the exact same experimental settings (e.g., input scale). We elaborate on the detailed experimental settings and the implications of the results below.

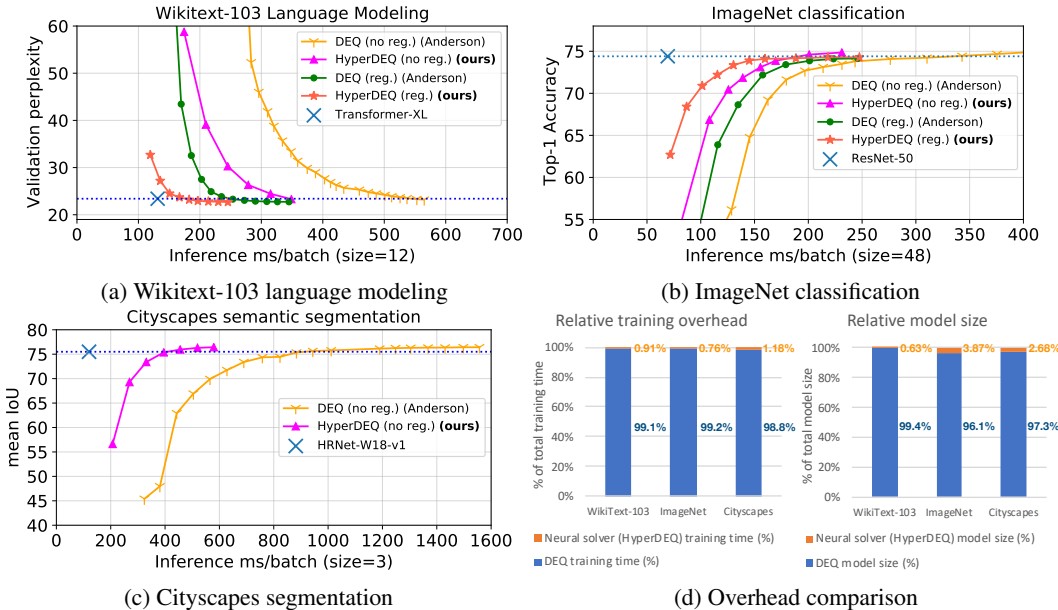

(a) Wikitext-103 language modeling

(b) ImageNet classification

(c) Cityscapes segmentation

(d) Overhead comparison

Figure 4: 4a- 4c: Comparisons of DEQs with classic and neural solvers. All speed/accuracy curves within the same plot are benchmarked on the same GPU with the same experimental setting, averaged over 6 independent runs. 4d: The overhead of DEQ hypersolver is extremely small.

**WikiText-103 Language Modeling.** In this experiment, $f_\theta$ is a Transformer layer (Vaswani et al., 2017; Dai et al., 2019; Bai et al., 2019) and the fixed points $\mathbf{z}^\star$ are (embeddings of) text sequences. We train the neural solver on sequences of length 60 for 5000 steps, and demonstrate its inference-time effect in Figure 4a (where we use a validation sequence length of 150). Specifically, compared with the original DEQ-Transformer (Bai et al., 2019) (Y curve), which uses generic Anderson acceleration (Anderson, 1965) or Broyden's method (Broyden, 1965) (both have similar pareto curves; see App. C), this same DEQ model solved with our neural approach (dubbed HyperDEQ; see ▲ curve) achieves significantly better efficiency. Moreover, our method is complementary to prior work that builds faster implicit models by Jacobian regularizations (Finlay et al., 2020; Bai et al., 2021). To demonstrate this, we additionally train a DEQ-Transformer model with Jacobian regularization (Bai et al., 2021) (● curve), and apply the neural solver on this regularized DEQ (★ curve). This movement of the speed/perplexity curves validates the DEQ property at the core of this paper: *the decoupling of the representational capacity (i.e., $f_\theta$) and the forward computation (i.e., the solver)*. With everything combined, we bring the performance of implicit Transformer-based DEQs close to the explicit Transformer-XL (Dai et al., 2019), which is the SOTA architecture on this task.

**ImageNet classification.** We additionally evaluate HyperDEQ on ImageNet classification ($224 \times 224$ images), customizing a neural solver on top of a 4-resolutional multiscale DEQ models (Bai et al., 2020). We train the HyperDEQ with 12 HyperAnderson iterations, and the speed/accuracy curves are shown in Figure 4b (▲ and ★ curves). Note that while Jacobian regularization (● curve) eventually hurts the performance of a multiscale DEQ (cf. Y curve) due to the strong constraint it imposes, the DEQ model with neural solver achieves faster inference without sacrifing any accuracy (since $f_\theta$, and thus $\mathbf{z}^\star$, are identical); e.g., we reach 75.0% accuracy while being almost $2\times$ faster.

**Cityscapes semantic segmentation.** We also show that our neural solver approach works well in domains where existing regularization-based methods (see Sec. 2) fail. Specifically, we apply the neural equilibrium solver on Cityscapes semantic segmentation, where the task objective is to label every pixel on a high-resolution (typically $2048 \times 1024$) image with the class of the object that the pixel belongs to. As in the ImageNet and WikiText-103 tasks, we found that there is a consistent gain in using the neural solver over the generic alternative, accelerating fixed-point convergence by more than a factor of 2 (see Figure 4c). In contrast, prior methods such as Jacobian regularization (Bai et al., 2021) do not work in this setting, due to their dependence on the exact structure of $f_\theta$. (Specifically, when $f_\theta$ is convolution-based and the image is very large, Jacobian regularization that encourages contractivity is at odds with the gradual broadening of the receptive field.) Our neural solver is orthogonal to the structure of $f_\theta$ (which is frozen), and we only improve how the solver functions.

## 5.2 Training Efficiency of the Neural Solver

We also provide extra training analysis in Fig. 4d. Not only is our approach effective, but the overhead for training the neural solver is also extremely small: the neural solver module is tiny ($< 4\%$ of the DEQ model size) and requires only about 1% of the training time needed by the original DEQ model (e.g., on WikiText-103, a DEQ requires 130 hours on 4 GPUs; the neural solver requires only about 1.2 extra hours). We believe this is strong evidence that neural solvers are simple, lightweight, and effective tools that take advantage of the decoupling properties of equilibrium models to yield an almost-free acceleration at inference time. We also perform convergence analysis in App. D.

Interestingly, one can also employ the neural solver to accelerate the DEQ training, but with three caveats: 1) during training the fixed point manifold also keeps changing; 2) we want to amortize the cost of computing "groudtruth" $\mathbf{z}^\star$; and 3) we still keep the backward implicit differentiation intact. Thus, we propose to train the neural solver $\{h_\phi, s_\xi\}$ and the DEQ model $f_\theta$ in an alternating manner, and elaborate more in App. D. We empirically observe this leads to a 16-20% DEQ training speedup.

## 5.3 Ablative Studies and Limitations

Finally, we perform a series of ablation studies to understand the benefits of multiple components within our design of the neural equilibrium solvers. We use the language modeling task on WikiText-103 for this purpose (where $f_\theta$ is a Transformer layer), while noting that we've noticed similar trends in all other settings. The results are presented in Fig. 5. The HyperDEQ with everything combined (initializer, $\alpha^k$, and $\beta_k$ predictions) performs best. Making the Anderson iterations learnable generally improves convergence. Moreover, although simply adding an initializer to a generic solver (✚ curve) does not help much, learning and backpropagating through the HyperAnderson iterations makes the initializer quite useful (cf. ■ and ⋆ curves). We additionally take the learned initializer $h_\phi$ from HyperDEQ and verify that this tiny module is by itself *still a poor language model* (see Table 1 and Sec. 4.3), but is valuable to our HyperAnderson iterations. More ablation studies (e.g., how $\alpha$ is predicted) are reported in Appendix C.

Table 1: Perplexity (ppl) on WikiText-103

| | Model Size | Test ppl |
|---|---|---|
| Gated ConvNet (Dauphin et al., 2017) | 230M | 37.2 |
| Transformer-XL (Dupont et al., 2019) | 165M | 24.2 |
| HyperDEQ (reg.) w/ 12 iters (**ours**) | 98M | **23.4** |
| Initializer $h_\phi$ (Conv1d) | 0.4M | 836.94 |

Figure 5: Ablative studies on HyperDEQ (reg.).

We also note two caveats for our approach. First, as mentioned in Sec. 4.3, backpropagating through the HyperAnderson iterations means the memory could grow with the number of steps $K$ that we run for. However, we don't find this to be problematic in practice, as we observed the training of these hypersolvers to be very insensitive to batch size , and that at inference time hypersolvers *do* easily generalize to iterations $> K$ (see also App. D). Second, though our method brings consistent improvements over generic solvers, in some cases a certain amount of iterations may still be required for good performance (e.g., $f_\theta$ is a $3 \times 3$ convolution and the input is a large image).

## 6 Discussion

We introduce a neural fixed-point solver for deep equilibrium (DEQ) models. The approach is simple, customizable, and extremely lightweight. Unlike prior works that regularize the structures or parameterizations of the implicit layer design (usually at the cost of accuracy), we propose to exploit this valuable notion of how implicit models decouple the representation (i.e., $f_\theta$) from the forwards computation. We directly learn a *model-specific* equilibrium solver that provides: 1) better-informed initial guesses; and 2) parameterized iterations that generalize Anderson acceleration and take into account future steps. Our experiments show that these modifications substantially improve the speed/accuracy trade-off across diverse large-scale tasks, while adding almost no overhead to training. We see these encouraging results as a significant step towards making implicit models more practical, and hope that this work will further motivate the application of implicit models such as Neural ODEs, DEQs, and other variants (Gu et al., 2020; Wang et al., 2020) to real, large-scale datasets.

## 7 ACKNOWLEDGEMENT

Shaojie Bai is supported by a grant from the Bosch Center for Artificial Intelligence (BCAI). We thank Brandon Amos for the helpful discussions.

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

Table 2: Task settings (see Sec. 5). Note that in WikiText-103 and ImageNet, we train the neural solver only for a few thousand gradient steps, which is far less than a complete epoch on these datasets. [†]In addition, we decay the loss weight $\lambda_3$ (for $\mathcal{L}_\alpha$) to 5e-8 on a linear schedule over the first 1.5-2K training steps (see below).

| Task | **Language modeling** | **Image classification** | **Semantic segmentation** |
|---|---|---|---|
| **Dataset** | WikiText-103 (Merity et al., 2017) | ImageNet (Deng et al., 2009) | Cityscapes (Cordts et al., 2016) |
| **Download link** | Link | Link | Link |
| **Split (train/val/test)** | 103M/218K/246K (words) | 1.28M/ - /150K (images) | 2975/500/1525 (images) |
| **Vocabulary size** | 267,735 | Not Applicable | Not Applicable |
| **Input type** | Text Sequence | Image | Image |
| **Implicit model arch.** | DEQ-Transformer | Multiscale-DEQ | Multiscale-DEQ |
| **Input scale (train)** | Length=60 | $H \times W$=224 × 224 | $H \times W$=2048 × 1024 |
| **Input scale (train)** | Length=150 | $H \times W$=224 × 224 | $H \times W$=1024 × 512 |
| $J_g(\mathbf{z}^\star)$ **size (per sample)** | $(1.05 \cdot 10^5) \times (1.05 \cdot 10^5)$ | $(1.88 \cdot 10^5) \times (1.88 \cdot 10^5)$ | $(7.86 \cdot 10^6) \times (7.86 \cdot 10^6)$ |
| **Batch size (train)** | 16 | 32 | 8 |
| **Optimizer (lr)** | Adam (0.001) | Adam (0.001) | Adam (0.001) |
| **HyperAnderson** $K$ **(train)** | 10 | 10 | 12 |
| **HyperAnderson storage** $m$ | 5 | 5 | 5 |
| **Training steps** $T$ | 5000 | 4000 | 3000 |
| **Loss weight** $\lambda_1$ **(for** $\mathcal{L}_{\mathbf{conv}}$**)** | 0.1 | 0.1 | 0.1 |
| **Loss weight** $\lambda_2$ **(for** $\mathcal{L}_{\mathbf{init}}$**)** | 0.05 | 0.02 | 0.02 |
| [†]**Loss weight** $\lambda_3$ **(for** $\mathcal{L}_\alpha$**)** | 1e-4 | 1e-5 | 1e-5 |

## A EXPERIMENTAL DETAILS

We describe here (and summarize in Table 2) the datasets, pretrained models and the training settings (including hyperparameters) for the experimental results reported in Sec. 5. We note that overall, since our approach is orthogonal to how these DEQ models were trained in the first place, we directly downloaded the pretrained DEQ models from the publicly released DEQ repo (Bai et al., 2019) (for language modeling) and the MDEQ repo (Bai et al., 2020) (for image classification and segmentation), which our code base is built upon. All of our experiments were conducted on NVIDIA RTX 2080 Ti GPUs. The pareto efficiencies were benchmarked on 1 GPU, averaged over 6 independently trained hypersolvers (i.e., 6 random seeds) for each task.

### A.1 WIKITEXT-103 WORD-LEVEL LANGUAGE MODELING

In a language modeling task, an autoregressive network is trained to predict the next token given a sequence of the past ones. Large-scale contextualized language models have been widely studied in literature, and is the cornerstone of many recent progress in realistic natural language AI models; e.g., ELMO (Peters et al., 2018), BERT (Devlin et al., 2019), XLNet (Yang et al., 2019), and GPT-3 (Brown et al., 2020). Formally, given an autoregressive network $F$ (like a Transformer (Vaswani et al., 2017) or LSTMs (Hochreiter & Schmidhuber, 1997)) and input sequence $\mathbf{x}_{1:t}$, the prediction $\hat{\mathbf{y}}_{1:t} = F(\mathbf{x}_{1:t}) \in \mathbb{R}^{t \times N}$ should be ideally as close as possible to the groundtruth "next words" $\mathbf{x}_{2:t+1}$, with $\mathbf{y}_t \in \mathbb{R}^N$ (i.e., like an $N$-way classification, where $N$ is the vocabulary size). Our experiments follow the exact same settings as the *DEQ-Transformer* network in Bai et al. (2019), where a deep equilibrium (DEQ) model is as a sequence model and $f_\theta$ a single Transformer block (i.e., multi-head self-attention (Vaswani et al., 2017; Dai et al., 2019)); i.e.,

$$\hat{\mathbf{y}}_{1:t} = h(\mathbf{z}_{1:t}^\star) \text{ where } \mathbf{z}_{1:t}^\star = f_\theta(\mathbf{z}_{1:t}^\star, \mathbf{x}_{1:t}), \text{ and } h \text{ a linear layer.} \tag{5}$$

We use one of the most commonly-used and the largest-scale textual corpus, Wikitext-103 (Merity et al., 2017), to evaluate our method. This dataset can be downloaded at this link. Specifically, the Wikitext-103 corpus contains over 103M words in its training split, and 218K/246K words for validation/test. Moreover, the entire corpus contains $N = 267,735$ unique words, which retain rare words, numbers, punctuation and case (as opposed to all text being lowercased) from the original Wikipedia articles.

Note that our approach only introduces minimal new hyperparameters (as the original DEQ model parameters are frozen). For the language modeling task, we use Adam optimizer (Kingma & Ba, 2015) with start learning rate 0.001 and cosine learning rate annealing (Loshchilov & Hutter, 2017). The neural solver is trained for 5000 steps, with sequences of length 60 and batch size 10, on top of

a pretrained DEQ with word embedding dimension 700. We set the HyperAnderson iteration limit $K$ to 10 (i.e., the HyperAnderson module is unrolled for 10 steps at training), while evaluating the hypersolver at validation time for up to $K = 16$ steps for the pareto efficiency curve benchmarking (see Fig. 4a).

## A.2 IMAGENET CLASSIFICATION

In both the Imagenet classification and the Cityscapes segmentation tasks (see below), we use a pretrained multiscale-DEQ (MDEQ-small, with 4 resolutions) model (Bai et al., 2020) for training our neural equilibrium solvers. In this case, feature maps maintained at multiple (e.g., $n$) resolutions are driven to their equilibria simultaneously; i.e.,

$$\mathbf{z}^\star = [z_1^\star, \ldots, z_n^\star] = f_\theta([z_1^\star, \ldots, z_n^\star], \mathbf{x}) \tag{6}$$

To verify our approach, we directly train and test on the largest vision datasets trained and reported for implicit models. The ImageNet dataset (Deng et al., 2009) contains 128,1280 training images and over 150K test images of resolution $224 \times 224$, which are distributed over 1,000 classes. Notably, as the DEQ model is already trained and frozen, we do *not* perform data augmentations (e.g., random cropping or horizontal flipping, which are standard practice for training models) to the input images, and make sure that the train/test inputs undergo the exact same transformations. The neural solver is trained for 4000 steps, each with 10 HyperAnderson iterations.

## A.3 CITYSCAPES SEGMENTATION

The other, even higher-resolution, vision dataset we apply the hypersolver on is the Cityscapes dataset (Cordts et al., 2016). This is a large-scale dataset that contains a diverse set of high-quality stereo video sequences recorded in street scenes from 50 different cities (at different times of the day, spanning several months), with pixel-level annotations of 5,000 frames over 19 classes (e.g., road, person, rider, truck, bus, etc.). Specifically, these 5,000 frames are divided into 2,975 training, 500 validation and 1,525 test images. The multiscale-DEQ (MDEQ) we use follow the exact formulation as in the ImageNet case above, with 4 resolutions driven to the fixed-point simultaneously.

The Cityscapes pixel-level semantic segmentation task is also a standard widely-used benchmark (Chen et al., 2018a; Wang et al., 2020; Cheng et al., 2020), and is especially challenging because of its rich contents and high resolution. To train the neural solver, as in ImageNet, we do not perform data augmentations like horizontal flipping (but perform random cropping to $1024 \times 512$) on the training set, where we set the HyperAnderson step limit to $K = 12$ and the batch size to 6. The neural solver is then evaluated in the standard full-resolution setting, on Cityscapes `val` $2048 \times 1024$ images.

## A.4 GROUNDTRUTH SOLVERS AND LOSS WEIGHTS

In all of the experiments, we use the canonical Broyden's method (Broyden, 1965) with 60 standard quasi-Newton steps as the "groundtruth" solver that yields the exact fixed point solution. However, overall we found the choice of this high-precision solver not important as long as we run it for enough number of steps to ensure proper convergence (e.g., a canonical Anderson acceleration algorithm with 60 steps would also suffice, since it's the same groundtruth fixed point).

The major hyperparameters introduced by the neural solver are the relative loss weights. For the fixed-point convergence loss $\mathcal{L}_{\text{conv}} = \sum_{k=1}^{K} w_k \|\mathbf{z}^{[k]} - \mathbf{z}^\star\|_2$ (see Sec. 4), we apply weights $w_k > 0$ to all intermediate HyperAnderson steps, with $w_k$ monotonically increasing (such as later HyperAnderson steps are applied a heavier penalty for deviation from the fixed point) and sum to 1; e.g., we can simply do

$$w_k = \frac{k}{\sum_{i=1}^{K} i} \tag{7}$$

Note that other similar formulations are possible, such as $w_k = \frac{k^2}{\sum_{i=1}^{K} i^2}$, and we generally do not find them empirically make a huge difference. We further illustrate this in the ablation studies presented in Appendix C.

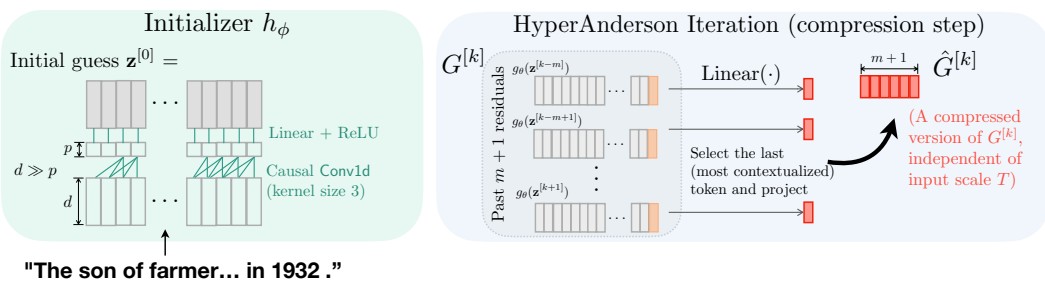

(a) Visualization of DEQ-Transformer (for WikiText-103 language modeling) initializer and HyperAnderson compression module.

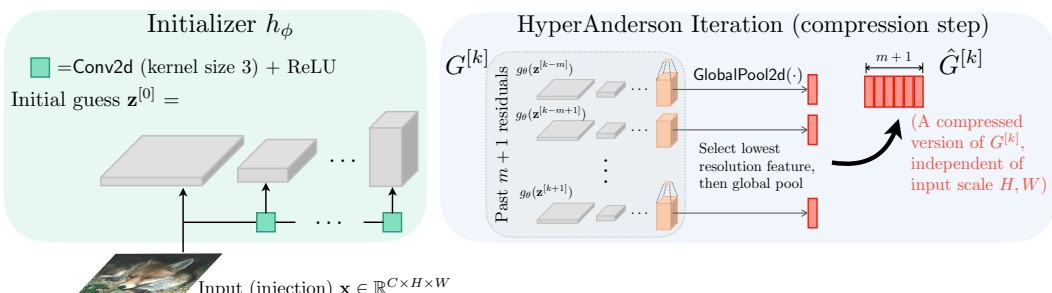

(b) Visualization of multiscale-DEQ (for ImageNet classifications and Cityscapes segmentations) initializer and HyperAnderson compression module.

Figure 6: Visualization of the neural equilibrium solver design. We use a local context and shallow layer to build the initializer, and perform global pooling over the input (1D sequences or 2D images) to create compressed (but representative) versions of the iterate $G^{[k]}$.

We applied a simple grid search on the relative loss weights $\lambda_1, \lambda_2$ and $\lambda_3$ on the three losses, finding it overall beneficial to adjust these weights so that the 3 losses at the start of the training be at roughly the same magnitude/scale. Moreover, we empirically find it useful to set $\lambda_3$ larger initially to encourage Anderson-like iterations (cf. Eq. equation 2), but later gradually decay this weight to almost 0, so that the hypersolver can more flexibly exploit the $\alpha^k$ in each HyperAnderson iterations. In all experiments, we decay $\lambda_3$ to 5e-8 on a linear schedule over 2,000 steps.

# B  ADDITIONAL DISCUSSIONS ON THE NEURAL EQULIBRIUM SOLVER DESIGN

We briefly illustrate in this section the designs of the initializer and the HyperAnderson iteration, and how we make them both lightweight and expressive (of the feature map). This implies two design principles that affect the model we use: 1) the hypersolver module itself should have an as minimal number of parameters as possible, which implies projections to low-dimensions; 2) it should be also invariant to input dimensionality while being able to cover the entire input. The second point is especially important since the original implicit model can be applied on inputs of various scales (e.g., sequences of length, 100, 200, etc.; same for images), and so it is preferable that the neural solver is also adaptive to such variety. We note that this rules out the learning-to-learn (L2L; see Sec. 2) way of parameterizing optmizers by RNNs: whereas the parameter space has a fixed dimensionality, hidden unit space does not.

**HyperDEQ for sequences.**   In the sequence modeling task, as mentioned in Sec. 4, our initializer $h_\phi$ is simply an 1D causal convolution followed by a non-linear projection (see Fig. 6a):

$$h_\phi(\mathbf{x}) = \text{ReLU}(\text{Conv1d}_{k=3}(\mathbf{x}))W$$

$$\text{where} \quad \text{Conv1d}_{k=3} : \mathbb{R}^{T \times d} \to \mathbb{R}^{T \times p} \qquad \text{(1D } \textit{causal} \text{ convolution with kernel size 3)}$$

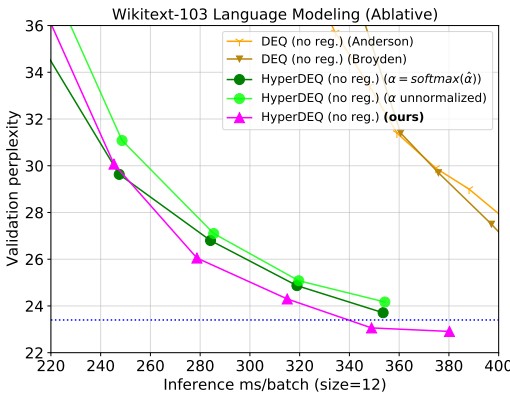
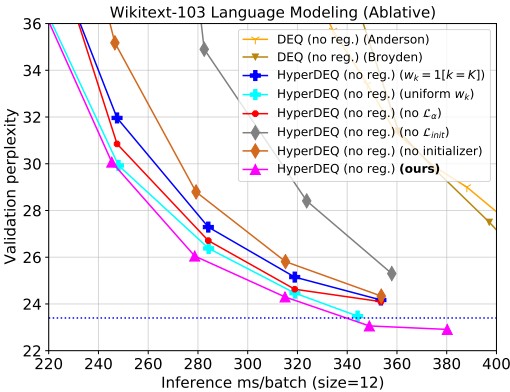

(a) Ablations on the importance of $\alpha$ normalization and learning. ▲ curve is our proposed method.

(b) Ablations on loss components and relative loss weights. ▲ curve is our proposed method.

Figure 7: Further ablations on the neural solver design (in terms of the $\alpha$ prediction and loss components. Note that we use an unregularized DEQ here (in contrast to Fig. 5) to better demonstrate the curve differences, which could sometimes be small.)

In particular, we set $p \ll d$ (e.g., in the WikiText-103 experiment, $d = 700$ and $p = 100$) so that this module requires little parameters and is fast to evaluate. We reiterate here that this is possible because the goal of the initializer is not to solve the original problem (e.g., high-dimensional language modeling), but only to give a "reasonable" initial guess to the fixed point solver. In other words, the initial guess itself may be bad in a task perspective (as we verified in Table 1) but good in an optimization perspective (see Fig. 5).

Moreover, we compress the past and current residuals $G^{[k]} = [g_\theta(\mathbf{z}^{[k-m]}), \ldots, g_\theta(\mathbf{z}^{[k+1]})] \in \mathbb{R}^{(m+1) \times T \times q}$ by simply taking the last and most contextualized token of each sequence; i.e., $(g_\theta(\mathbf{z}^{[i]}))_T \in \mathbb{R}^q$. This is reasonable because the design of a sequence model like Transformer layers (which parameterizes our $f_\theta$ for DEQ) already ensures that the last token contains most of the contextual information of the entire sequence (and this technique is frequently used for textual classification tasks, or in vision transformers (Dosovitskiy et al., 2020)). We therefore propose to compress the past residuals of the fixed point estimates by directly using the (a projection of the) very last tokens (see Fig. 6a), which are collected and treated as a mini time-series as mentioned in Sec. 4.

**HyperDEQ (multiscale) for vision.** The initializer for multiscale DEQ models is similarly designed, with a series of small-context 1-layer convolutions (e.g., kernel size 3) to generate the initial guess for *each* resolution stream (see Fig. 6b). To compress the past residuals, we leverage the fact that all resolutions are fused in each layer of the multiscale DEQ design, and only take the lowest-resolution feature map $z_n^{[i]}$ of the past fixed point estimate $\mathbf{z}^{[i]} = [z_1^{[i]}, \ldots, z_n^{[i]}]$ (in practice, we use $n = 4$). As shown in Fig. 6b, we directly apply global pooling to pass into the later HyperAnderson steps (by treating it as a mini time-series, like in the sequence case).

## C  ADDITIONAL EFFICIENCY AND ABLATIVE STUDIES

We further study some ablative settings not covered in Sec. 5.3 here. We specifically focus on two aspects: 1) how $\alpha$ is predicted; and 2) the effects of different loss components.

In the original AA prototype, the values of $\alpha$ is determined by solving a least-squared solution over the past $m$ steps, as described by the constrained optimization problem equation 2. In particular, these $\alpha^k = (\alpha_0^k, \ldots, \alpha_m^k)$ values can be any real value, but has to sum to 1. In our hypersolver, we propose to "normalize" the predicted $\hat{\alpha}^k$ by a shift: $\alpha^k = \hat{\alpha}^k + \frac{(1 - \mathbf{1}^\top \hat{\alpha}^k)}{m_k + 1} \cdot \mathbf{1}$. We here compare two alternatives: no normalization at all (i.e., have $\alpha$ learned freely; see ● in Fig. 7a), and softmax-based normalization (i.e., requiring $\alpha^k$ to be all-positive; see ● in Fig. 7a). Specifically, we find that the exact choice of $\alpha$ normalization does not affect the overall substantial pareto curve improvement (e.g., compare them with the ⅄ curve baseline), but ensuring that the $\alpha^k$ values can be negative while

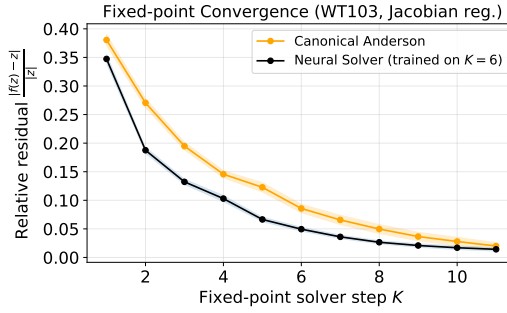 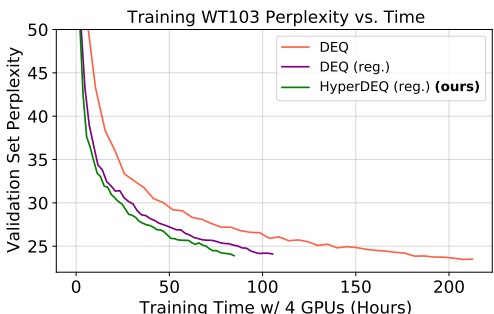

(a) The neural equilibrium solver improves upon the convergence path of canonical Anderson acceleration. The speedup of HyperDEQ is a result of this faster convergence *and* the more lightweight solver step.

(b) Hypersolver can also be used at training time to further accelerate the DEQ model training by alternating their training (in addition to existing methods like Jacobian stabilization).

Figure 8: Left: Convergence analysis of the hypersolver at inference time. Right: Although trained with a frozen $f_\theta$, the neural equilibrium solver can also be used to accelerate DEQ training.

still normalized (i.e., sum to 1) overall benefit the performance. For instance, with the same inference speed, we get a 1-1.5 perplexity improvement.

We also analyze the effect of different loss components in Fig. 7b. For the main fixed-point convergence loss $\mathcal{L}_{\text{conv}}$, we compare two alternatives of the relative weights applied on the intermediate HyperAnderson steps: only apply $w_k = 1$ at the final output $K$ (see ✚ curve), or set $w_k$ to be uniform for all $k$ (see ✚ curve). We empirically find that the hypersolver trained in both scenarios perform well, but the monotonically incresing $w_k$ we use (i.e., putting larger emphasis on later iterations) perform best. Moreover, removing either the initializer loss $\mathcal{L}_{\text{init}}$ (note that we still keep the initializer $h_\phi$ itself, just don't supervise it; see the ⧫) or the alpha loss (the ● curve) $\mathcal{L}_\alpha$ impacts the performance. Interestingly, removing the $\mathcal{L}_{\text{init}}$ loss on the initializer yields even worse performance than even removing the initializer itself (the ⧫ curve). However, we note that all of these ablative settings still significantly outperform the speed/accuracy efficiency than the generic solvers, which suggests the overall benefit of using customized learnable solvers for implicit models.

Finally, we provide additional empirical evidence on the convergence and generalizability of the neural solvers in Fig. 8a, where we compare the convergence of a pre-trained, regularized Transformer-based DEQ model under 1) canonical Anderson acceleration; and 2) a neural solver (with initializer) trained to unroll for $K = 6$ HyperAnderson steps. The shaded areas around the curves correspond to the standard deviation of the relative residual at each solver step, measured on 1000 randomly selected textual sequences of length 100. From Figure 8a, we first note that a hypersolver trained with $K$ unrolling steps is able to generalize (i.e., converge well) beyond $K$, although both solvers eventually plateau at some of $\varepsilon$ residual value. Second, with the parameterized fixed-point solving, we see that the neural solver consistently improves over the canonical solver's convergence, while being more lightweight to compute (i.e., each solver neural solver step is cheaper than canonical Anderson step), which account for the 2× speedup we observe across the board for DEQ model inference efficiency in Sec. 5.

## D    USE HYPERSOLVER FOR DEQ TRAINING

It turns out we could also leverage its acceleration power to improve the not just the inference but also training of the original DEQ model (i.e., replace the classical solver at DEQ training with a hypersolver). However, we note three major caveats. First, as the DEQ model is being trained (e.g., by SGD), parameterization of the layer $f_\theta$ also changes, which means the fixed-point manifold of this implicit model constantly gets updated. Therefore, a neural deep equilibrium solver $\{h_\phi, s_\xi\}$ trained on a parameter state $f_{\theta_0}$ may gradually deviate from accurate and stable convergence as $f_\theta$ also deviates from $f_{\theta_0}$ as a result of more and more parameter updates. Second, precisely computing the groundtruth fixed points $\mathbf{z}^\star$ with a classical solver at every training step would add heavy burden to the training overhead. Third, the usage of hypersolver to solve forward pass fixed point does not affect the backward pass implicit differentiation (notably, decoupled forward and backward trajectories is

also a major feature of implicit models), implying the training efficiency improvement will be much smaller than that at the inference time (about 2×, as mentioned in Sec. 5).

Despite these challenges, exploiting the fact that model parameters $\theta$ drift only gradually with training iterations (i.e., at training step $t$ and $t + 1$, we assume $\|\theta_t - \theta_{t+1}\|$ is small), we propose to train the neural solver $\{h_\phi, s_\xi\}$ (which are tiny, lightweight, and have little representational capacity) and the implicit layer $f_\theta$ (which are huge, representationally rich, and requires many training steps) in an alternating way. Specifically, we adopt the following procedure:

1. Warmup and train the DEQ model for some small number of steps.
2. Take a snapshot of the current implicit layer $f_\theta$ and train *from scratch* a hypersolver $\{h_\phi, s_\xi\}$ customized on top of it.
3. Use the current hypersolver $\{h_\phi, s_\xi\}$ to solve for the fixed point and train the DEQ model for the next $M$ steps.
4. Take a snapshot of the current implicit layer $f_\theta$, and *fine-tune* the existing hypersolver $\{h_\phi, s_\xi\}$ on top of it by minimizing $\mathcal{L}(\omega, K)$ for some $T$ hypersolver training steps.
5. Repeat step 3 & 4 until we reach max training steps for the DEQ model.

Empirically, we find that the proposed neural solver generalizes surprisingly well for values of $M$ that are not too large (i.e., even though $f_\theta$ gets updated, the hypersolver can still reliably converge to the correct fixed points during the next $M$ DEQ training steps). Moreover, as previously mentioned, training neural solver is extremely lightweight, and even more so for fine-tuning it, suggesting that $T$ is very small in pratice. For example, on WikiText-103 language modeling with a transformer-based $f_\theta$ (which has 300K DEQ training steps), we set $M = 5000$ and $T = 200$; in other words, we fine-tune the hypersolver for merely 200 steps (again, on very small batch size and sequence length; see Table 2) after every 5000 DEQ training steps, in order to keep the neural solver "up to date" with the implicit layer. This extra (but minor) cost of constantly fine-tuning hypersolver is counterbalanced by the huge gain in using hypersolver over a canonical generic root solver.

Fig. 8b shows the training efficiency improvement as a result of adopting hypersolver in DEQ training. With the hypersolver fine-tuned intermittently (green curve), we are able to reduce the total training time over that of the classical-solver-based DEQ counterpart by about 20% (purple curve). Note that just like at inference time, this improvement via hypersolver is orthogonal (and thus additional) to the acceleration effect of regularization-based methods, like Jacobian regularizations (cf. purple and pink curves).

## E    ADDITIONAL EXPERIMENT ON TINY EQUILIBRIUM MODELS

While Fig. 4 provides strong evidence on how the hypersolver could improve the speed/accuracy pareto efficiency of these large-scale DEQ models (e.g., DEQ-Transformer contains almost 100M parameters), we also perform additional experiments to validate its efficacy on much smaller networks. Specifically, we train a hypersolver on a tiny MDEQ model that is pretrained on CIFAR-10 classification task (the same setting as in Bai et al. (2020), but with Jacobian regularization) which contains only 170K parameters. The hypersolver contains about 8K parameters (including the initializer $h_\phi$), and is trained for only 1000 training steps. The results are shown in Fig. 9. The improvement in pareto frontier again suggests that the neural solvers provide a lightweight and yet effective solution to improving these implicit models.

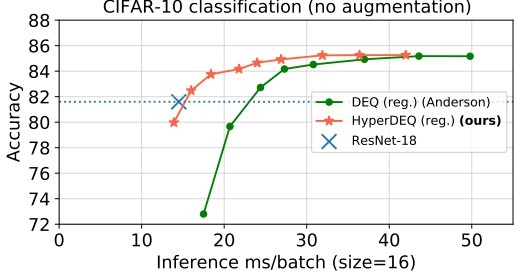

Figure 9: The benefit of hypersolver holds on tiny implicit models. Inference speed is benchmarked with batch size 16, averaged over 5 runs.

