# OpenReview forum: "Neural Deep Equilibrium Solvers"
_ICLR.cc/2022/Conference — ICLR 2022 Poster_

### Official Review · Reviewer_CYHx · 2021-11-01

**Correctness:** 4
**Technical Novelty And Significance:** 3
**Empirical Novelty And Significance:** 4
**Recommendation:** 8
**Confidence:** 4

**Main Review:**

[Strength] The paper has the following strength.
- The paper researches the very interesting, important, and difficult problem of accelerating implicit model inference. The paper gives a working solution using neural networks to assist the inference fixed point calculation with impressive results.
- The method offers significant speedup (2x) in forward steps over the current solvers for fixed point calcuation at inference. Bringing the inference speed of the model close to explicit forward feeding models.
- Thorough empirical analysis on the property of the solver is well conducted in the paper, showing the overhead for training the solver is minimal compared with the training of the implicit models. It is also demonstrated that the solver generalizes and scales well to large experiments in natrual language processing and computer vision.
- Further experiments on creative usage of the solver at training time shows that the merits of the speedup when incorporated in training is larger than the overhead on training the solver. This shows that the application of the method is "free".

[Weekness] The paper is well written but has a few glitches. I think the paper would be a good addition to ICLR if the authors can carefully address them.
- The notations of the paper has some typos. Is G^{[k]} in (2) a matrix? If not, the L2 norm can be defined clearly. Is the dimension of \hat{G}^{[k]} of dimension C by (m_k + 1) instead of (m_k + 1) by C?
- The use of Hyper Anderson Iterations can be more clearly justified. After reading the Sec. 4.1, I am still curious how much worse it would be to do the naive thing by solving problem (2) (maybe approximately). Explicit explanation with some numbers could help.
- The paper does not offer significant theroetical contribution but this is not the goal of the paper and the reveiwer believes that it should not undermine its empirical contributions.
- The merits of the proposed solver should be highly general and go beyond deep equilibrium models (DEMs) to other implicit models including implicit archetectures for graph-structured data (Gu, 2020) and implicit Feature Pyramid Network for object detection (Wang, 2020). Discussions and potentially experiments about how the method would work on them can only improve the merits of the work.

Ref:

Gu, F., Chang, H., Zhu, W., Sojoudi, S., El Ghaoui, L. (2020). Implicit graph neural networks. Advances in Neural Information Processing Systems 33, 11984--11995. https://papers.nips.cc/paper/2020/hash/8b5c8441a8ff8e151b191c53c1842a38-Abstract.html

Wang, T., Zhang, X., & Sun, J. (2020). Implicit feature pyramid network for object detection. arXiv preprint arXiv:2012.13563. https://arxiv.org/abs/2012.13563

**Summary Of The Paper:**

The paper presents a method called neural deep equilibrium solver to increase the efficiency in the inference stage for implicit deep models by initializing the equilibrium states using neural network. The authors start with the traditional Anderson Acceleration scheme for fixed point calculation and extend it using neural network initialization and Anderson steps to improve the inference efficiency. The authors conduct comprehensive experiments to demonstrate that the speed up in inference is significant and general with little overhead at training time. The further experiments shows that the proposed method can be incorporated in the training procedure to give faster training while introducing the speedup at inference time.

**Summary Of The Review:**

This paper presents a method to significantly improve the inference of implicit models. Although the paper does not focus on theoretical contribution, it demonstrates empirical merits very well and the results are impressive. The paper is written clearly and is a solid work overall.

---

> ### Author Response · Authors · 2021-11-19
> **Response to reviewer CYHx**
>
> Thank you for your positive feedback and the questions! We address them below.
>
> > Q: Is $G^{[k]}$ in (2) a matrix? If not, the L2 norm can be defined clearly. Is the dimension of $\hat{G}^{[k]}$ of dimension C by (m_k + 1) instead of (m_k + 1) by C?
>
> Thank you for pointing this out! Yes, the dimension of $G^{[k]}$ is indeed $C \times (m_k+1)$ (e.g., we compute $\|G^{[k]}\alpha^k\|_2$ in the alpha loss $\mathcal{L}_\alpha$), and it is a matrix, since it collects the past $m_k+1$ residuals (which we flatten into vectors). We have updated the draft to correct this.
>
> > Q: The use of Hyper Anderson Iterations can be more clearly justified... I am still curious how much worse it would be to do the naive thing by solving problem (2) (maybe approximately). Explicit explanation with some numbers could help.
>
> We first note that this neural solver is still an Anderson-based solver, which completely fits the AA prototype (see Alg. 1 in our paper). We mainly replace the mechanism for determining the mixing procedure (just like the type-I and type-II AA variants) to be now content-based and trainable. One major justification behind this is the fact that the underlying dynamical system of a DEQ model is defined by both the layer $f_\theta$ and the input $\mathbf{x}$. In other words, given a frozen DEQ model, for a different input $\mathbf{x}$ the dynamical system will be different also. This implies that we can probably perform **input-dependent** convergence, which is exactly what these HyperAnderson iterations are doing (e.g., we produce initial guess based on $\mathbf{x}$; we train them with BPTT; we customize it toward a frozen $f_\theta$). This is in contrast to generic solvers (e.g., the canonical AA that solves Eq. (2)), which is designed for general convergence and does not account for the fact that this convergence may be input-dependent as well.
>
> Solving Eq. (2) definitely still works, as it simply reduces to the canonical Anderson acceleration algorithm. To see (quantiatively) how much difference the HyperAnderson iteration makes, we refer the reviewer to Figure 8(a) in Appendix C, where we explicitly compare the convergences. Both solvers work well, but the HyperAnderson iterations not only 1) converge faster overall; but 2) are also more lightweight (because we do not need to solve Eq.(2), and the $\alpha_k$ can be generated by a tiny neural network). These two factors together contribute to the improvement in pareto curves we see in Figure 4.
>
> > Q: The paper does not offer significant theoretical contribution but this is not the goal of the paper and the reviewer believes that it should not undermine its empirical contributions.
>
> We agree with the reviewer that the efficacy of the proposed neural solver is empirically validated and that it’d be interesting for future work to look into some of the theoretical properties of these neural high-dimensional fixed-point solvers.
>
> > Q: The merits of the proposed solver should be highly general and go beyond deep equilibrium models (DEMs) to other implicit models including implicit archetectures for graph-structured data (Gu, 2020) and implicit Feature Pyramid Network for object detection (Wang, 2020).
>
> We completely agree with the reviewer, and from the empirical evidence that we gathered so far on large DEQ models, we do believe hypersolvers point to a broader set of applications that could extend beyond the scope of deep equilibrium networks to other implicit networks. As we mentioned in the paper as well, the hypersolver is generally not predicated on the underlying $f_\theta$ function (e.g., we used Transformer blocks and multiscale residual blocks). Since both IGNNs and iFPNs are convergent fixed-point-based models (e.g., iFPN also uses Broyden’s method especially), we agree with the reviewer and expect that they will likely get a similar boost in pareto efficiency indeed. We have cited these (and other related) papers in our revised draft, and are currently looking into them. We’d like to thank the reviewer for pointing this out and are glad to include more results and analysis in the paper.

---

### Official Review · Reviewer_eZVG · 2021-11-01

**Correctness:** 3
**Technical Novelty And Significance:** 2
**Empirical Novelty And Significance:** 4
**Recommendation:** 8
**Confidence:** 4

**Main Review:**

Strengths:
- The technique is intuitively motivated as a learnable version of Anderson acceleration, which while lacking theoretical basis, is easy to follow and a natural method to try.
- The experimental evaluation is convincing. The extensive empirical evaluation of the proposed approach gives the reader the sense that the hypersolver is at least worth trying on their DEQ model.

Weaknesses:
1. Being a largely empirically/intuitively motivated work, there are many user settings and hyperparameters (and even hardcoded arbitrary choices, e.g. the 3 losses introduced in section 4.2) that have to be chosen within any theoretical guidance and are not well-motivated. Still, this work opens the door for future mathemtical analysis of the proposed method, although perhaps on a more restricted or smaller practical/empirical scale.
2. In L_conv, is z^\ast necessarily a true solution, or just one provided by another solver? None of the losses in section 4.2 actually minimise the ostensible goal of the equilibrium solver. Why is there no loss that directly minimises the absolute value of the residual |g|? It seems like the neural solver is learning to imitate the behaviour of a provided solver (given access to z^\ast, which may not actually be a root, nor does a root necessarily exist, nor is it guaranteed to be unique), rather than solve the problem directly. Did you try to minimise |g| instead? Any reason or intuition as to why you chose this method?

Comments/suggestions/questions:

3. One of the really nice features of your technique is that (as you mention) the fixed point solver does not need training labels in order to update its parameters. This ability to do unsupervised training could be highlighted in the abstract.
4. Typo. "...we treat the it as a mini time-series of length..."
5. Typo. "...2) these hyprtsolvers can be trained very quickly..."
6. Figure 4d. All of the interesting part of the graph occupies 99.6% to 100%. Remove the bottom 99.6% of the vertical axis. Maybe even a logarithmic scale is appropriate.
7. It may be present in the appendix but I was not able to easily find it. How many random seeds do the curves in Figure 4 represent? (hopefully not one...!) Is each point independent from each other (does each point use different random seeds), both within the same colour and outside of the same colour?
8. Can you try your method on some smaller networks (e.g. around 1,000-100,000 parameters) and simpler tasks to see if an advantage over Anderson acceleration still persists? I am interested in understanding whether the success of this method is tied to the difficulty of the fixed point problem, which should intuitively be greater in larger models.


**Summary Of The Paper:**

The authors introduce a neural network approach for solving the fixed point equations arising in deep equilibrium models. This consists of a tiny network that provides an initial guess for the fixed point, as well as a small network that computes coefficients inside an algorithm inspired by Anderson iteration. The approach is intuitive and empirical. Although no theory is given, the authors demonstrate the strength of their proposed solver in large scale experimental evaluations. Specifically, the new solver is fast to train, has a small parameter count, and appears to drastically shift the pareto front of the inference speed/performance curve for all DEQ models.

**Summary Of The Review:**

An empirically motivated neural network replacing the role of a traditional root finder (e.g. Anderson acceleration) in DEQ models appears to improve performance and inference time. However, this root finding network introduces a new set of hyperparameters and all the baggage usually associated with deep learning (no theory of convergence, optimisation, generalisation, hyperparameter choice) (weakness 1), and it is not clear whether the choices made by the authors are universally applicable (weakness 1,2). All in all, a good paper that is likely to be adopted by the community and spark future research.

---

> ### Author Response · Authors · 2021-11-19
> **Response to reviewer eZVG**
>
> Thank you for your valuable feedback! We are happy to discuss more and address your questions as follows.
>
> > Q: There are many user settings and hyperparameters (and even hardcoded arbitrary choices, e.g. the 3 losses introduced in section 4.2) that have to be chosen within any theoretical guidance and are not well-motivated. Still, this work opens the door for future mathemtical analysis of the proposed method, although perhaps on a more restricted or smaller practical/empirical scale.
>
> We agree with the reviewer that the fact that this hypersolver is based on another (albeit small) neural network makes obtaining theoretical insights and guarantees challenging. On the one hand, we do rely on this tiny neural solver to learn by itself to approach the fixed point (e.g., $\mathcal{L}_\text{conv}$ and $\mathcal{L}_\text{init}$); but on the other hand, we also want to highlight that this hypersolver is still an AA-based solver, which completely fits the AA prototype (see Alg. 1 in our paper). This in fact (at least partly) explains why our approach needs very minimal training (as it’s still AA); and why its iterations can generalize beyond the BPTT length $K$ it was trained for.
>
> We also share the reviewer’s point that this could open the door to stricter theoretical analysis of the efficacy of these (high-dimensional) hyper fixed-point solvers, which has many potential applications outside the scope of DEQ models (e.g., in scientific computing). A potential caveat though, is that the neural solver here is customized towards a particular high-dimensional function $f_\theta$ and is not proposed as a generic solver, which might limit the scope of this mathematical analysis.
>
> > Q: Is $z^\star$ necessarily a true solution, or just one provided by another solver? Did you try to minimise $\|g_\theta(\mathbf{z}; \mathbf{x})\|$ instead? Any reason or intuition as to why you chose this method?
>
> $\mathbf{z}^\star$ is the solution provided by another (generic) solver, like Broyden’s method, which we run for a large number of iterations (e.g., 70 steps) to ensure it’s quite close to the real fixed point. On the residual loss -- great question! In fact, we did try stepwise residual loss $\mathcal{L}_\text{res} = \sum_i \frac{1}{C} \|f_\theta(\mathbf{z}^{[i]}; \mathbf{x}) - \mathbf{z}^{[i]}\|_2$ when we first worked on this project (where $C$ is a normalization term; e.g., it could be the initial residual). We have two observations:
> 1. Without the $\mathcal{L}_\text{conv}$ loss, this residual loss indeed helps train the hypersolver. We are still able to improve the pareto frontier of the DEQ model at inference time, but the improvements are substantially weaker than if we use $\mathcal{L}_\text{conv}$.
> 2. With the presence of $\mathcal{L}_\text{conv}$, we find that adding $\mathcal{L}_\text{res}$ does not help improve the hypersolver at all, likely because $\mathcal{L}_\text{conv}$ already provides a good fixed-point target that the BPTT can optimize for.
>
> Moreover, although we are using $\mathbf{z}^\star$, we note that the training of hypersolver is still also **completely unsupervised**, in the sense that we are not using any “ground truth” label. Due to these empirical reasons and observations, we removed the residual loss from our design.
>
>
> > Q: How many random seeds do the curves in Figure 4 represent? (hopefully not one...!) Is each point independent from each other (does each point use different random seeds), both within the same colour and outside of the same colour?
>
> For the plots in Figure 4, we use the same underlying pretrained DEQ model (e.g., we can download https://github.com/locuslab/deq; we also train some of them on our own). Then, for each of these DEQ models, the pareto curves are computed by averaging over 6 independently trained hypersolvers (i.e., 6 random seeds). The points on the curve simply correspond to the discrete neural solver steps (e.g., we unroll each hypersolver for k=1, 2, …, K, … steps) and the inference time is averaged over all batches in the validation set, and over the 6 hypersolvers. Points from different colors, though, are from different hypersolvers (as the $f_\theta$ functions are different). However, we note that empirically we observe very little variance over these hypersolvers even with different seeds; they almost always have very similar levels of performance after about 5K training steps. We will include this information in the caption of Figure 4 and in Appendix A.

---

> > ### Author Response · Authors · 2021-11-19
> > **Response to reviewer eZVG (part 2)**
> >
> > > Q: Can you try your method on some smaller networks (e.g. around 1,000-100,000 parameters) and simpler tasks to see if an advantage over Anderson acceleration still persists?
> >
> > Following the reviewer’s advice, during the rebuttal we tried to train a hypersolver on a tiny multiscale DEQ model (MDEQ) which contains only 170K parameters and has been used on CIFAR-10 (without data augmentation) [1]. We report below the result (inference speed and accuracy) of the HyperAnderson vs. canonical AA iterations, where our neural solver only used ~8K parameters (including the initializer) and was trained for 1000 steps with BPTT length K=5:
> >
> > | Solvers   | Iter 4 (Acc/speed) | 5       | 6         | 7         | 8         | 10        | 12        |
> > | ------------ | ----------------------- | ------- | -------- | --------- | -------- | --------- | ---------- |
> > | Anderson      | 72.80 (17.5ms)     | 79.67 (20.7ms) | 82.72 (24.4ms) | 84.17 (27.3ms) | 84.52 (30.8ms) | 84.93 (37.0ms) | 85.18 (43.6ms) |
> > | Neural Solver | 82.48 (16.0ms)     | 83.75 (18.4ms) | 84.17 (21.8ms) | 84.66 (24.0ms) | 84.92 (26.9ms) | 85.24 (31.9ms) | 85.26 (36.4ms) |
> >
> > We benchmark the avg. inference speed with a batch size of 16, averaged over 5 runs. As we can see, even on this much smaller network it still benefits to use a hypersolver, which converges much faster (i.e., better performance with same # of steps) and is more lightweight (i.e., each step is cheaper) than a generic solver. We have included and visualized these results in the (new) Appendix E of the paper as well.
> >
> > [1] https://arxiv.org/pdf/2006.08656
> >
> >
> > We also thank the reviewer for pointing out the typos and the suggestion on the abstract to include unsupervised training. We have fixed and included these discussions in the revised draft!

---

### Official Review · Reviewer_YWYD · 2021-11-03

**Correctness:** 4
**Technical Novelty And Significance:** 4
**Empirical Novelty And Significance:** 4
**Recommendation:** 8
**Confidence:** 4

**Main Review:**

The introduction, motivation, and overview of DEQs and their contribution is very strong. They provide extensive experimentation proving the benefits of their method across many large-scale tasks, pushing the state of the art of DEQs closer to practical deployment.

Some questions:
- Does the learned extension of AA still guarantee fixed-point convergence? Is it enough to enforce that the $\alpha_k$ weights sum to $1$?
- Thanks to this paper, DEQ models seem to be approaching practicality during inference time. How does their training time compare to explicit models? (I understand this is orthogonal to this paper. I'm just curious.)
- One caveat you mention is that BPTT to train HyperAnderson network could use a lot of memory. I think this could be substantially reduced with gradient checkpointing, if it ever becomes a bottleneck.

**Summary Of The Paper:**

The paper proposes to speed up inference of Deep Equilibrium Models (DEQs) by replacing the classic fixed-point solvers (Broyden or Anderson Acceleration) by a learned extension of AA. Their approach operates on a pre-trained DEQ, and trains a small neural network to propose an initialization and update scheme based on ground truth fixed points. Their method is orthogonal to existing regularization approaches to speeding up DEQs.

The paper has extensive experiments across large scale tasks: Language modeling, ImageNet classification, and Semantic Segmentation. They show pareto improvements across these tasks as compared to standard DEQs, while only adding a ~1% additional training overhead of the DEQ.

**Summary Of The Review:**

I believe this is a very strong paper that significantly pushes the state of the art of DEQs, and should get accepted without reservation (barring something substantial I may have missed).

---

> ### Author Response · Authors · 2021-11-19
> **Response to reviewer YWYD**
>
> Thank you for your positive evaluation and questions. We are also excited about the prospect that DEQ models are closing their inference-time efficiency gap with explicit models! We answer your questions as follows:
>
> > Q: Does the learned extension of AA still guarantee fixed-point convergence? Is it enough to enforce that the $\alpha_k$ weights sum to 1?
>
> Our empirical analysis (on a large amount of high-dimensional data) showed that the neural solver is indeed able to converge in the right direction to the right fixed point quite consistently, and to generalize beyond the steps $K$ it was trained for.  There are multiple reasons for this, of which two are most important.
> 1. This neural solver is still an AA-based solver, which fits the AA prototype (see Alg. 1 in our paper), where we mainly replace the mechanism for determining mixing procedure (just like the type-I and type-II AA variants) to be now content-based and trainable. Therefore, besides ensuring that $\alpha_k$’s sum to 1 as the reviewer mentioned, we still need to ensure that these mixing weights are acting in the right direction for each hyperAnderson step, which is what $\mathcal{L}_\alpha$ is doing.
> 2. We apply this at the inference time DEQ models, which means the fixed-point manifold is by itself stable and fixed, and so is the model expressivity.
>
> However, as the hypersolver is still a neural network and is updated by SGD, it would be challenging to prove (or guarantee) its convergence property.
>
> > Q: Thanks to this paper, DEQ models seem to be approaching practicality during inference time. How does their training time compare to explicit models?
>
> In their canonical form, these equilibrium networks are about 3-4x slower than explicit models to train (e.g., see [1,2]), because 1) both the forward and backward passes still involve iteratively solving a high-dimensional fixed-point system; and 2) these models (as well as neural ODEs) tend to get more and more unstable during training, which make their training iterations more and more costly. However, we do note that there are some recent followup works like [2] and [3] that show that we could train these models with Jacobian-free backward pass (which makes the backward pass extremely cheap), or regularize the model to be more stable (at a small cost to generalization performances).
>
> [1] https://arxiv.org/pdf/1909.01377.pdf
>
> [2] https://arxiv.org/pdf/2106.14342.pdf
>
> [3] https://arxiv.org/pdf/2103.12803.pdf
>
> > Q: One caveat you mention is that BPTT to train HyperAnderson network could use a lot of memory. I think this could be substantially reduced with gradient checkpointing, if it ever becomes a bottleneck.
>
> We completely agree with the reviewer that gradient checkpointing will be a very useful technique to help train these hypersolvers with longer BPTT chains! We will update this discussion in the revised draft.

---

> > ### Comment · Reviewer_YWYD · 2021-11-30
> > **Response to response**
> >
> > Thank you for the response to my review! I keep my decision that this is a strong paper that should be accepted to ICLR.

---

### Decision · Program_Chairs · 2022-01-20

**Decision:**

Accept (Poster)

**Comment:**

The authors introduce a neural network approach for solving the fixed point equations arising in deep equilibrium models. This consists of a tiny network that provides an initial guess for the fixed point, as well as a small network that computes coefficients inside an algorithm inspired by Anderson iteration.

Overall, there is consensus among the reviewers that the paper is well written and is a strong empirical study.

I recommend acceptance as a poster.

Additional remarks:

- The authors argue the DEQs / implicit deep learning models allow a decoupling between representational capacity and inference-time efficiency. Yet, in the "Regularizing Implicit Models" paragraph, they write "Implicit models are known to be slow during training and inference. To address this, recent works have developed certain regularization methods that encourage these models to be more stable and thus easier to solve.", which seems like a contradiction to me. So while in theory I agree with this decoupling, in practice, it seems not completely true.

- Section 3 should include some discussion on conditions on f_theta for the existence of a fixed point.

- Since the initialization and HyperAnderson networks are trained using unrolling, there is some memory overhead compared to vanilla DEQs, that are differentiated purely using implicit differentiation. It would be great to clarify the amount of extra memory needed by these networks. It is necessary to justify that the initialization and HyperAnderson networks are smaller than usual neural networks.